# Histone tails cooperate to control the breathing of genomic nucleosomes

**Jan Huertas**[1,2,3], **Hans Robert Schöler**[2,4], **Vlad Cojocaru**[1,2,3]*

**1** In Silico Biomolecular Structure and Dynamics Group, Hubrecht Institute, Utrecht, The Netherlands,
**2** Department of Cellular and Developmental Biology, Max Planck Institute for Molecular Biomedicine,
Münster, Germany, **3** Center for Multiscale Theory and Computation, Westfälische Wilhelms University,
Münster, Germany, **4** Medical Faculty, University of Münster, Münster, Germany

* v.cojocaru@hubrecht.eu, vlad.cojocaru@cojocarulab.eu

## Abstract

Genomic DNA is packaged in chromatin, a dynamic fiber variable in size and compaction. In chromatin, repeating nucleosome units wrap 145–147 DNA basepairs around histone proteins. Genetic and epigenetic regulation of genes relies on structural transitions in chromatin which are driven by intra- and inter-nucleosome dynamics and modulated by chemical modifications of the unstructured terminal tails of histones. Here we demonstrate how the interplay between histone H3 and H2A tails control ample nucleosome breathing motions. We monitored large openings of two genomic nucleosomes, and only moderate breathing of an engineered nucleosome in atomistic molecular simulations amounting to 24 μs. Transitions between open and closed nucleosome conformations were mediated by the displacement and changes in compaction of the two histone tails. These motions involved changes in the DNA interaction profiles of clusters of epigenetic regulatory aminoacids in the tails. Removing the histone tails resulted in a large increase of the amplitude of nucleosome breathing but did not change the sequence dependent pattern of the motions. Histone tail modulated nucleosome breathing is a key mechanism of chromatin dynamics with important implications for epigenetic regulation.

**Data Availability Statement:** All simulations included in this manuscript are available for download at the following link: https://datashare.mpcdf.mpg.de/s/kjegwpBEFUEidwR. The download is password protected. The password is:

## Author summary

In the cell, the DNA is packed in chromatin. Chromatin is a highly dynamic fiber structure made of arrays of nucleosomes with different degrees of compaction. Each nucleosome has 145–147 basepairs of DNA wrapped around a protein octamer made of four unique histone proteins. Each histone is present twice and has a structured part and one or two disordered terminal tails. The regulation of gene expression in the cell and during cellular transitions depends on dynamic changes in chromatin structure. Chromatin dynamics are modulated by intra and inter nucleosome motions and by posttranslational chemical modifications of the histone tails. Here we reveal how histone tails control the intra nucleosome dynamics at atomic resolution. From extensive sampling of nucleosome dynamics in atomistic molecular simulations, we show that genomic nucleosomes breath more extensively than engineered ones and we describe how two histone tails cooperate

"huertas2021!". A README file is available that describes the simulations.

**Funding:** This work was supported by funds of the Max Planck Society (J.H., H.S.) and The Royal Netherlands Academy of Arts and Sciences (V.C.). Computer resources were provided by the Gauss Centre for Supercomputing e.V. (www.gauss-centre.eu) (project ID 12622, STRUCNUCREC running on the GCS Supercomputer "SuperMUC" at the Leibniz Supercomputing Centre (www.lrz.de), awarded to J.H. and V.C.). The funders had no role in study design, data collection and analysis, decision to publish, or preparation of the manuscript.

**Competing interests:** The authors have declared that no competing interests exist.

to control nucleosome breathing through interactions between clusters of positively charged residues and the DNA. Nucleosome conformations with different degrees of opening are associated with different conformations, positions, and DNA interaction patterns of the tails. With this mechanism, we contribute to the understanding of chromatin dynamics at atomic resolution.

## Introduction

In eukaryotic cells, the DNA is packed into chromatin, a dynamic fiber structure made of arrays of nucleosomes. In each nucleosome, 145–147 basepairs of DNA are wrapped around a protein octamer made of 4 histones (H3, H4, H2A, H2B), each present twice [1, 2]. The N-terminus regions of all histones and the C-terminus of H2A (H2AC) are disordered, positively charged tails, carrying ∼28% of the mass of the structured core of the histones [1].

Histone tails interact with the DNA in a non-specific manner, protruding from the nucleosome superhelix and embracing the DNA. Their role in controlling the gene expression has been extensively analyzed [3–5]. H3 and H4 tails mediate the interaction between nucleosomes [6, 7], while the H2AC tail affects nucleosome dynamics, repositioning, and interacts with proteins modulating chromatin dynamics, such as linker histones or chromatin remodelers [8]. The charged residues of histone tails are targets for post-translational modifications which are added chemical groups that act as epigenetic markers to regulate chromatin accessibility and gene expression at a given time in a specific cellular context [3]. For example, the tri-methylation of K9 [9] or K27 in H3 [10] mark regions of closed, inactive chromatin, contributing to gene silencing. Contrarily, lysine acetylation in H3 [11] and H4 [12] mark open, actively transcribed regions of chromatin. These tails modulate chromatin structure by controlling both inter- and intra-nucleosome interactions.

The impact of the tails on mononucleosome structural dynamics has been investigated with high resolution techniques such as single-molecule Förster Ressonance Energy Transfer. Such experiments confirmed the role of the H3 tail as a "close pin" that modulates nucleosome breathing and unwrapping [13]. However, characterizing histone tail structure and dynamics at atomic resolution is difficult. Only one experimentally determined structure of nucleosomes has the histone tails resolved [14].

Nucleosome structure and dynamics at atomic resolution can be measured from molecular dynamics (MD) simulations. In these, time traces of the molecular system are generated by solving Newton's equation of motion. Because of the long time scale of large amplitude nucleosome motions and the bulky, disordered nature of the histone tails, the computational resources required for atomistic simulations are prohibitive. For this reason, many simulations were performed with simplified, coarse grained nucleosomes [15–22]. From these, the role of DNA sequence in nucleosome unwrapping [17, 19, 20], protein-mediated nucleosome remodeling [22], or the impact of histone tails on nucleosome mobility, intra-nucleosome interactions and nucleosome unwrapping [15, 18, 21, 23] were evaluated. However, the simplifications used in coarse grained models hinder the accurate characterization of essential physical atomic interactions. Therefore, for a complete characterization of how histone tails interact with DNA and control nucleosome dynamics all-atom MD simulations are indispensable. Such simulations have been mostly performed with engineered or incomplete nucleosomes and the sampling achieved was limited [24]. For example, from biased MD simulations of nucleosomes without histone tails, the histone-DNA interactions involved in nucleosome unwrapping were determined [25]. Nucleosome opening has been observed at high salt

concentrations when the tails were partially truncated [26]. The hydration patterns around the nucleosome core [27] and the local flexibility of nucleosomal DNA have been characterized [28–30]. However, these studies provide only a partial description of nucleosome dynamics due to the absence of histone tails.

The structural flexibility of the tails was demonstrated in simulations of free histones or tail mimicking peptides [31, 32]. In the few atomistic simulations of complete nucleosomes reported to date, the structural flexibility was limited due to the short time scales or the small number of trajectories [33–39]. For example, limited nucleosome breathing and the collapse of histone tails on DNA was observed in a few 1$\mu$s simulations [36]. The mechanisms by which H2A histone variants alter nucleosome dynamics were proposed from an ensemble of 4 600 ns long simulations [37]. From a 3.36$\mu$s ensemble of $\sim$ 100 ns short trajectories at high temperature (353 K), opening of truncated nucleosomes with either the H3 or H2AC tails removed was observed [38].

These studies have been performed using mainly two DNA sequences: a palindromic sequence derived from human $\alpha$-satellite DNA present in the crystal structure with tails [14], and the Widom 601 sequence, an artificial sequence selected for strong positioning and high stability of the nucleosomes [40] also present in several crystal structures [41, 42]. For gene regulation in cells, the structural flexibility and mobility of nucleosomes with genomic DNA sequences matters most: genomic nucleosomes are more dynamic than engineered ones [43]. Recently, we characterized the breathing and twisting motions of two genomic nucleosomes from MD simulations as part of our effort to elucidate how the master regulator of stem cell pluripotency Oct4 binds to DNA wrapped in nucleosomes [39]. The two genomic nucleosomes bear regulatory sequences for the genes *ESRRB* and *LIN28B*, which are important for stem cell pluripotency [44, 45]. We showed that the motions of the Lin28b nucleosome are more ample than those of the strongly positioned engineered Widom nucleosome. Both the Lin28b and Esrrb nucleosomes were shown to be bound by the pioneer transcription factor Oct4 when converting skin to stem cells from a superposition of the nucleosome positioning with Oct4 binding data [46, 47]. We used these data to build models of the two genomic nucleosomes [39] and validated the models by monitoring the accessibility of the known Oct4 binding sites.

Here, from a total of 24 $\mu$s atomistic simulations of the engineered Widom and the genomic Lin28b and Esrrb nucleosomes, including the already reported 9 $\mu$s performed with *Drosophila* histones (from now on dH simulations) and additional 9 $\mu$s with human histones (hH simulations), 3 $\mu$s with tailless nucleosomes, and 3 $\mu$s with selected DNA and histone tail conformations, we demonstrate how the interplay between histone H3 and H2AC tails controls nucleosome breathing motions. The tails mainly kept the nucleosomes closed but allowed short lived transitions to extensively open conformations in the genomic nucleosomes. These were observed only when the tails were in configurations with few or no interactions with the outer DNA gyre. Because the distribution of open and closed nucleosomes impacts the structure and compaction of chromatin fibers, the mechanism we describe is key to understanding chromatin dynamics.

## Methods

### Nucleosome modeling

The 146 bp Widom nucleosomes with *Drosophila melanogaster* and *Homo Sapiens* histones (S1 Document) were built by homology modelling, using Modeller (https://salilab.org/modeller/), with the same procedure described previously [39]. As templates, we used the structure of the *Drosophila melanogaster* nucleosome core (PDB ID: 2PYO), the structure of

the Widom 601 nucleosome particle (PDB: 3LZ0), and the crystal structure of the X. laevis nucleosome that includes histone tails (PDB: 1KX5). For each nucleosome, 100 homology models were generated using a "slow" optimization protocol and a "slow" MD-based refinement protocol as defined in Modeller. We selected the lowest energy model and added an 11 bp fragment of B-DNA (generated with Nucleic Acid Builder from Ambertools 18 [48]) to each linker DNA to generate the 168 bp nucleosomes.

The Esrrb and Lin28b regulatory sequences were the same as used in our previous work [39] (S2 Document), originally selected from data by Soufi et al [46]. We substituted the Widom sequence with the Lin28b and the Esrrb sequences using the "swapna" function in Chimera (https://www.cgl.ucsf.edu/chimera/). The position of the dyad was chosen to be at the center of the Mononuclease digestion peak presented by Soufi et al [46]. In the human Lin28b nucleosome, the position was shifted by 1 extra bp to the 5' end compared to the *Drosophila* Lin28b. This shift was introduced following our previous work [39] to optimize the accessibility of the Oct4 binding sites.

The tail-less nucleosomes (TL) were obtained by removing all histone tails from the minimized structures of the human Widom, human Esrrb and *Drosophila* Lin28b nucleosomes. These nucleosomes are identical between the two species. Esrrb-TL and Lin28b-TL were modeled starting from those complete nucleosomes that showed the largest openings.

## Molecular dynamics simulations

The MD protocol was described previously [39]. In short, each nucleosome was solvated in a truncated octahedron box of at least 12 Å of SPCE water molecules around the solute in any direction. $Na^+$ ions were added to neutralize the charge of the system, and $K^+$ and $Cl^-$ ions were added up to a concentration of 150mM KCl. Then an energy minimization using the AMBER software [48] was performed, followed by 10.25 ns of equilibration with gradually decreasing positonal and base pairing restraints [39, 49]. The equilibration and the following production runs were performed using NAMD [50]. The production runs were in the isobaric-isothermic (NPT, p = 1 atm T = 300 K) ensemble, using Langevin dynamics as a termostat and the Nose-Hoover/Langevin piston method for pressure control. The AMBER force field parameters ff14SB [51], parmbsc1 [52] and Li-Merz [53] were used for protein, DNA and ions, respectively. We performed three independent simulations, each 1 $\mu$s long, for each complete nucleosome, and a single 1 $\mu$s long simulation for each TL nucleosome.

## Analysis of nucleosome dynamics

All simulations were fitted with a root mean-square fit of the heavy atoms of the histone core to the minimized structure of the Widom nucleosome. The histone core was defined by excluding the histone tails: residues 1–45 for hH3 and dH3, 1–32 for hH4 and dH4, 1–18 and 129 for hH2A, 1–17 and 115–124 for dH2A, and 1–33 for hH2B and 1–31 for dH2B.

The characterization of the breathing motions was done using the procedure first described by Öztürk et al. [54], that we also used in our previous work [39]. We first defined a coordinate system *XYZ* with the origin on the dyad. *X* was defined as the vector along the dyad axis.*Y* was defined as the cross product between *X* and a vector perpendicular to *X* intersecting it approximately at the center of the nucleosome. Finally *Z* was defined as the cross product between *X* and *Y*. Then the vectors $v_3$ and $v_5$ were defined along the 3' and 5' L-DNAs. The angle $\gamma_1$ was defined as the angle between the projection of these vectors in the *XZ* plane and the *Z* axis, whereas $\gamma_2$ was defined as the angle between the projection of the vectors on the *XY* plane and the *Y* axis.

The radius of gyration of the DNA ($R_g$) and the number of protein-DNA contacts were calculated using the cpptraj software [55]. Contacts were defined between non-hydrogen atoms closer than 4.5 Å. We split the DNA in two parts. The two outer gyres were defined as 40 bp from each end of the nucleosomal DNA, and the inner gyre was defined to include all remaining base pairs. For the histones, contacts were split between contacts made by the histone core and the histone tails, using the core and tail definition described above.

Mean and minimal distances ($\delta$ and $\delta_{min}$) were measured using the weighted mean distance collective variable *distanceInv* implemented in the Colvars module [56] available in VMD [57]. The weighted distance between two groups of atoms is defined as follows:

$$d_{1,2}^{[n]} = \left( \frac{1}{N_1 N_2} \sum_{i,j} \left( \frac{1}{||d^{ij}||} \right)^n \right)^{-1/n},$$ where $||d^{ij}||$ is the distance between atoms $i$ and $j$ in groups 1 and 2 respectively, and $n$ is an even integer. This distance will asymptotically approach the minimal distance when increasing $n$. For smaller values of $n$, weighted distance will be close to the true average distance. For measuring the position of the histone tails relative to DNA, we calculated the weighted mean distances with $n = 10$ ($\delta$, from now on referred to as "mean distances") from the tails to two reference points: the L-DNA, defined as the last 15 basepairs of each DNA arm, and the dyad, defined as the 8 bp centered on the dyad on the same side as the tail (3' and 5'). Minimal distances ($\delta_{min}$) between individual residues in histone tails and the inner gyre or the outer gyre of DNA (as defined above), were calculated as weighted mean distances with $n = 100$.

The $R_g$ and the $\gamma$ angles were calculated every 20 ps (in S1 Data every 40 ps) whereas the $N_C$ and the $\delta$ and $\delta_{min}$ every 100 ps (S2 Data).

## Clustering of MD trajectories

MD trajectories of all nucleosomes with the same histones were combined, obtaining 2 groups of simulations: one with human histones (hH simulations) and one with *Drosophila* histones (dH simulations). hH and dH simulations were separated because a clustering of the combined ensemble of simulations based on the histone tails would not be possible due to the differences in histone sequences between the 2 organisms. The snapshots from each group of simulations were clustered based on the positional RMSD of the backbone heavy atoms of DNA, H3 and H2A. For the DNA, the outer gyre that didn't display big opening events (5' and 3' in hH and dH simulations respectively) was not considered in the clustering. For the histone H3, the RMSD of the entire histone was used for clustering, whereas for the H2A, the RMSD of the entire histone excluding the N-terminal tail was used. This resulted in 3 independent clustering distributions, referred to as "DNA clusters", "H3 clusters", and "H2AC clusters". The k-means clustering algorithm implemented in cpptraj was used, with 8 centroids. Finally, the percentage of snapshots from each DNA cluster in each H3 and H2AC cluster was computed, obtaining the distributions of the DNA cluster snapshots in the histone tail clusters to evaluate the correlation between DNA conformations and the H3 and H2AC conformations and positions.

## Results

### Breathing motions are more extensive in genomic nucleosomes

We performed 3 independent 1 $\mu$s long simulations for each combination of DNA and histones (Table 1), which we grouped in ensembles of 3 $\mu$s for analysis. The highest structural flexibility we monitored in the L-DNA arms (both 5' and 3') and the histone tails (Fig 1), shown by the broader range of the number of contacts with DNA formed by the histone tails compared to the histone core (Table 1 and S1 Fig). All tails interacted with the core

**Table 1. Overview of the simulations performed.**

| Histones | DNA | Replica | Time | $R_g$[1] | Number of contacts[2] with DNA | | |
| --- | --- | --- | --- | --- | --- | --- | --- |
| | | | | | Histone Core | Histone Tails | |
| | | | | | | inner gyre[3] | outer gyre[3] |
| *Drosophila* | Widom | 1 | 1 $\mu$s | 47.11–48.34 | 1914–2124 | 790–1514 | 891–1360 |
| *Drosophila* | Widom | 2 | 1 $\mu$s | 47.62–49.58 | 1837–2151 | 683–982 | 743–1250 |
| *Drosophila* | Widom | 3 | 1 $\mu$s | 48.08–49.03 | 1780–2019 | 769–1299 | 907–1421 |
| *Drosophila* | Esrrb | 1 | 1 $\mu$s | 47.37–48.82 | 2010–2265 | 937–1388 | 764–1187 |
| *Drosophila* | Esrrb | 2 | 1 $\mu$s | 47.26–48.34 | 1869–2161 | 675–1269 | 989–1295 |
| *Drosophila* | Esrrb | 3 | 1 $\mu$s | 47.11–48.41 | 2013–2241 | 894–1375 | 783–1146 |
| *Drosophila* | Lin28b | 1 | 1 $\mu$s | 48.00–49.91 | 1917–2170 | 837–1353 | 942–1354 |
| *Drosophila* | Lin28b | 2 | 1 $\mu$s | 48.34–50.90 | 1928–2173 | 755–1718 | 639–1185 |
| *Drosophila* | Lin28b | 3 | 1 $\mu$s | 47.63–49.26 | 1977–2171 | 917–1384 | 812–1066 |
| Human | Widom | 1 | 1 $\mu$s | 47.93–49.26 | 2017–2224 | 762–1260 | 754–1031 |
| Human | Widom | 2 | 1 $\mu$s | 47.26–48.54 | 1867–2116 | 1061–1673 | 717–1023 |
| Human | Widom | 3 | 1 $\mu$s | 48.06–49.18 | 1807–2047 | 856–1442 | 716–1226 |
| Human | Esrrb | 1 | 1 $\mu$s | 47.83–51.32 | 1708–2057 | 1147–1893 | 457–763 |
| Human | Esrrb | 2 | 1 $\mu$s | 48.22–49.87 | 1844–2062 | 888–1631 | 777–1184 |
| Human | Esrrb | 3 | 1 $\mu$s | 47.14–48.18 | 1926–2134 | 901–1438 | 666–1098 |
| Human | Lin28b | 1 | 1 $\mu$s | 47.79–48.93 | 1907–2121 | 752–1123 | 828–1488 |
| Human | Lin28b | 2 | 1 $\mu$s | 47.93–49.00 | 1852–2048 | 827–1284 | 830–1261 |
| Human | Lin28b | 3 | 1 $\mu$s | 47.90–49.62 | 1792–2048 | 667–1549 | 739–1119 |
| Tail-less[4] | Widom | 1 | 1 $\mu$s | 48.48–53.22 | 1754–2038 | – | – |
| Tail-less[4] | Esrrb | 1 | 1 $\mu$s | 51.03–55.59 | 1808–2058 | – | – |
| Tail-less[4] | Lin28b | 1 | 1 $\mu$s | 48.87–54.96 | 1845–2117 | – | – |

[1] $R_g$ and the number of contacts are shown as the percentiles 5 to 95.

[2] A contact was defined between two non-hydrogen atoms closer than 4.5 Å to another non-hydrogen atom.

[3] Inner and outer gyre of the nucleosomal DNA, see Methods for details.

[4] Widom-TL", "Esrrb-TL", and "Lin28b-TL" were obtained by removing the histone tails from the human Widom, human Esrrb, and *Drosophila* Lin28b respectively.

nucleosomal DNA (146 bp centered on the dyad), but only the H3 and H2AC tails interacted also with the linker DNA (L-DNA).

To characterize structural flexibility, we computed the radius of gyration ($R_g$) and two angles, $\gamma_1$ and $\gamma_2$ (time series in S1 Data). These provide a complete description of nucleosome breathing in the XZ ($\gamma_1$) and XY ($\gamma_2$) planes defined by a dyad-centered coordinate system [39, 54] (Fig 2A, see Methods). The main $R_g$ peaks of the nucleosomes with human and *Drosophila* histones (hH and dH respectively) overlap, indicating a similar conformational space sampling in the 2 simulation sets. The genomic nucleosomes Lin28b and Esrrb adopted more open conformations with the exception of the Esrrb$^{dH}$ nucleosome (Fig 2B and S2 Fig). In contrast, both Widom nucleosomes sampled mainly conformations with $R_g$ around $\approx$ 48.25 Å and less often conformations with $R_g \approx$ 47.5 Å. The latter was similar with the closed conformation sampled by Esrrb$^{dH}$. The Esrrb$^{hH}$ nucleosome sampled three regions of the conformational space, two of them have $R_g$ values resembling those sampled by the Widom nucleosomes, while the third has more open conformations ($R_g > 50$). The Lin28b nucleosomes were more open ($R_g$ 48.25—49.5 Å) and did not sample the most closed conformations. Additionally, the Lin28b$^{dH}$ sampled open conformations with $R_g > 50$ Å, similar to those sampled by Esrrb$^{hH}$ (Fig 2B).

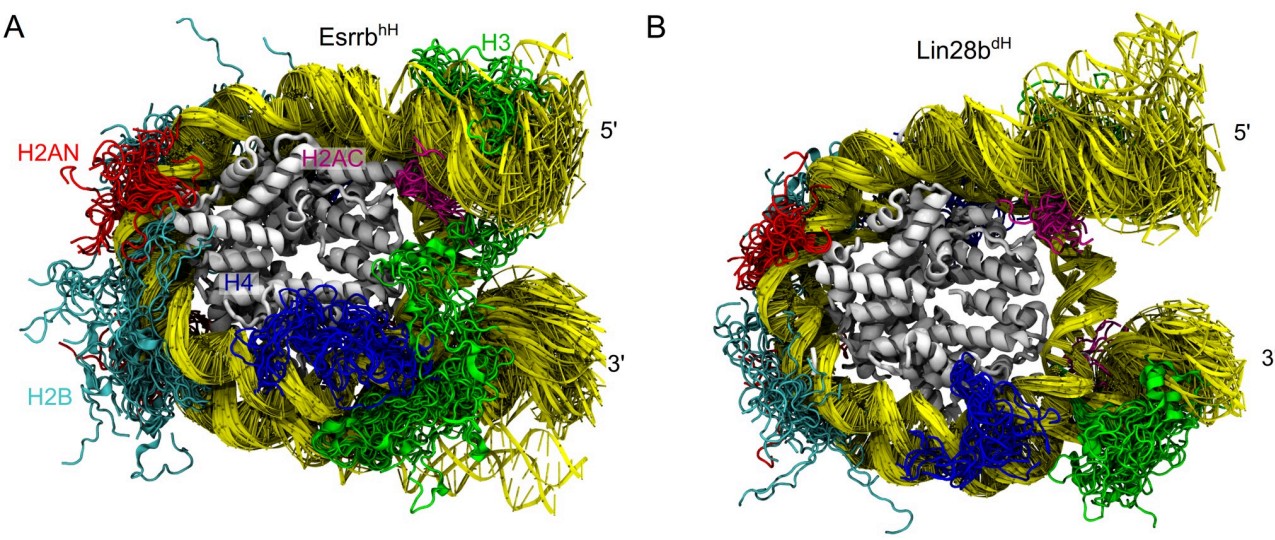

**Fig 1. Nucleosome motions on the *μs* time scale.** Superposition of nucleosome snapshots taken from the 3 *μs* simulation ensembles in which the major opening events occurred. (A) Esrrb^hH (B) Lin28b^dH. Tails are shown every 100ns, the DNA every 50 ns. The histone core is in gray, DNA in yellow. The histone tails of H3, H4, H2A (both N and C terminal) and H2B are in green, blue, red, magenta and cyan respectively.

In summary, the genomic nucleosomes breath more extensively than engineered nucleosomes on *μs* time scale. We monitored 2 large amplitude transient openings, one at the 3' L-DNA of Esrrb^hH and one the 5' L-DNA of Lin28b^dH.

The open conformations of Esrrb^hH and Lin28b^dH with highest $R_g$ were sampled in simulations 2 and 1 of each ensemble respectively and were characterized by fewer contacts between the histone tails and the outer DNA gyre (Table 1 and S1 Fig). The opening was either in both XZ and XY planes, or only in one plane and was induced by motions of the 3' and 5' L-DNA in the two nucleosomes respectively (Fig 2C).

To achieve a more extensive sampling of the breathing motions and further probe the sequence dependence of the breathing, we removed the histone tails from each nucleosome. Without the tails, dH and hH are identical, thus we performed a single 1 micros simulation for each nucleosome. All three nucleosomes open more extensively in these simulations (Fig 3). Remarkably, the opening pattern was similar to that observed in the complete nucleosomes. The Widom-TL nucleosome opened with the smallest amplitude, whereas the Lin28b-TL and Esrrb-TL opened more extensively at the 5' and 3' end respectively. These findings indicate that the histone tails keep the nucleosome closed, while the pattern of nucleosome breathing is encoded in the DNA sequence.

### Histone H3 and H2A tails sample a broad range of configurations

To test how the nucleosome conformational plasticity and the positional fluctuations of the histone tails are correlated, we first analyzed the histone tail configurations, defined as the combination between the tail position and its conformation. We monitored the position of the H3 and H2AC tails relative to to the neighbouring L-DNA and the DNA segment around the dyad and the number of contacts of these tails with the inner and outer DNA gyre (see Methods). We focused on these two tails as they are the only tails near the L-DNAs.

We found that the H3 and H2AC tails neighboring the highly flexible L-DNA arms (3' L-DNA in Esrrb^hH and the 5' L-DNA in Lin28b^dH) sampled a broad range of configurations (Fig 4 and S3 Fig, complete data in S2 Data). Different configurations were sampled in

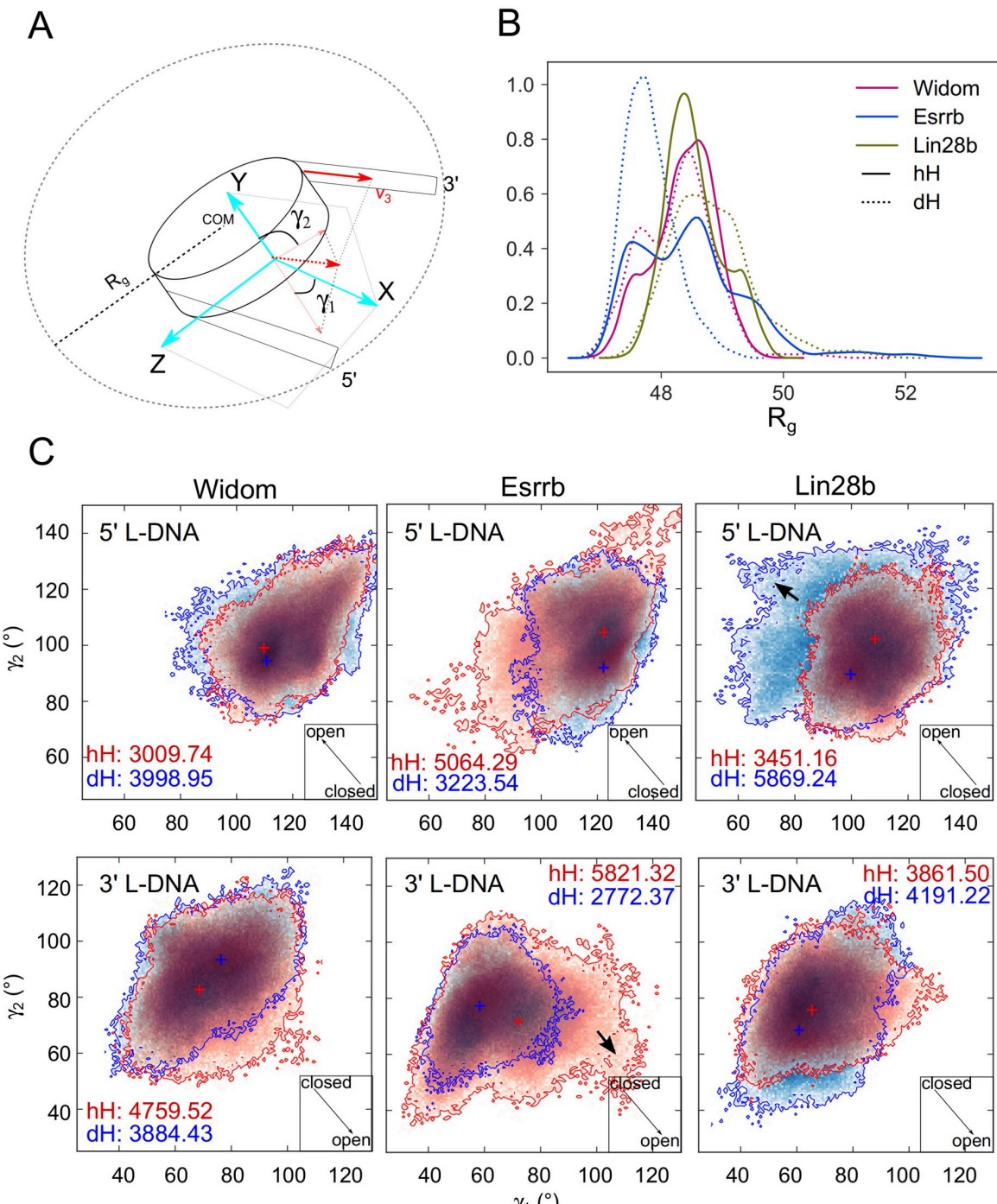

**Fig 2. Nucleosome structural flexibility.** (A) Schematic representation of the $R_g$ definition and the coordinate system used to describe nucleosome conformations. The angles $\gamma_1$ and $\gamma_2$, describing the nucleosome breathing motions were defined as described in Methods. An increase of $\gamma_1$ indicates opening at the 3' L-DNA, but closing at the 5' L-DNA, and vice-versa for $\gamma_2$. (B) $R_g$ distribution from the 3 $\mu$s ensembles of simulations. Larger values indicate a lower degree of compaction in the nucleosomal DNA. Solid and dashed lines represent the hH and dH simulations, respectively. The Widom, Esrrb, and Lin28b nucleosomes are shown in magenta, blue and green respectively. (C) Two-dimensional histograms depicting the conformational sampling of the 5' and 3' L-DNA arms in the space defined by the $\gamma_1$ and $\gamma_2$ angles. The arrow inserts in the lower-right corner indicate the direction of the opening, whereas the numbers are the areas of the histograms. The crosses indicate the most populated region of each histogram. The data for the hH and dH nucleosomes are in red and blue respectively. The two bolded arrows indicate the two extensive openings (complete data in S1 Data).

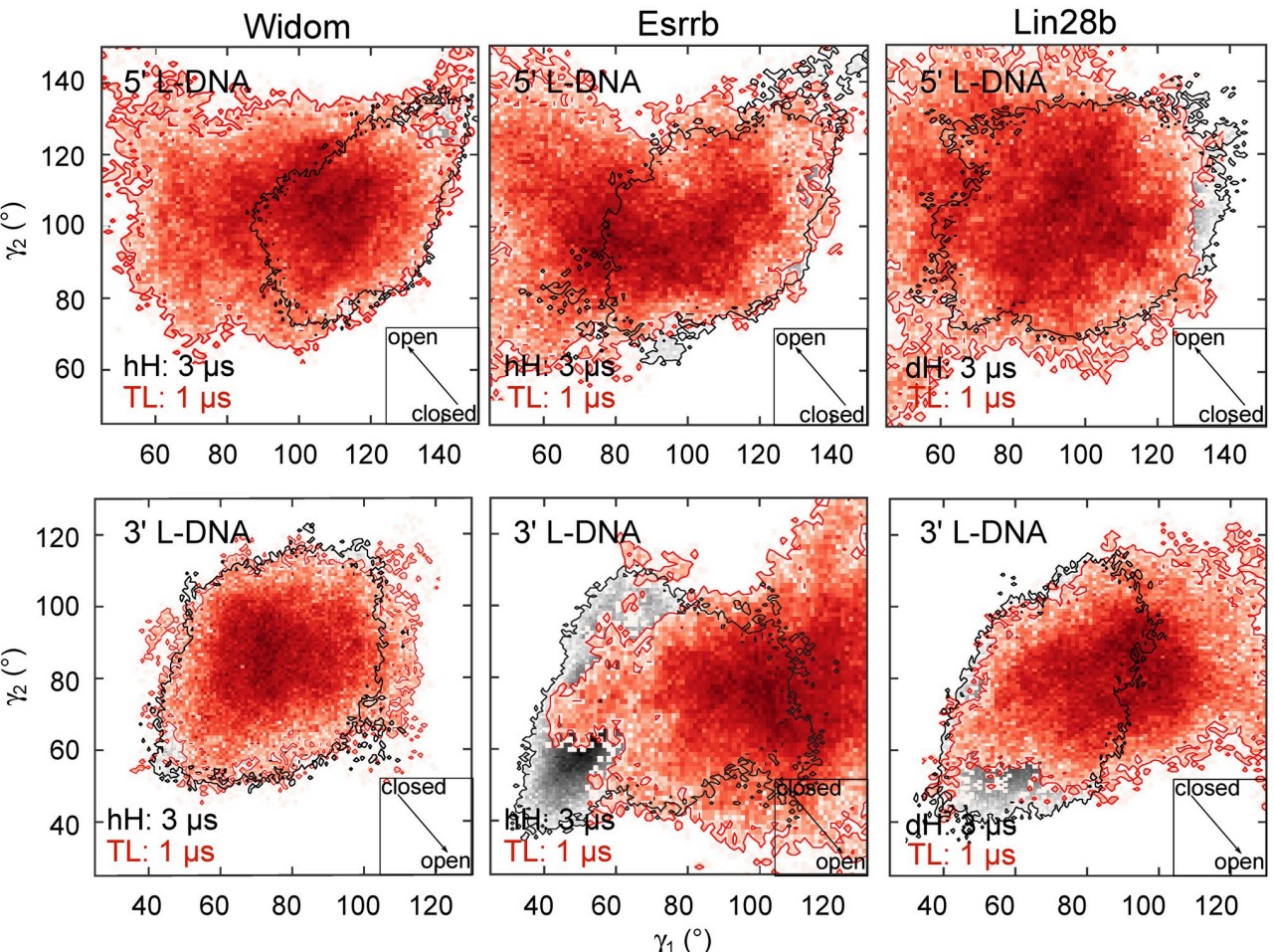

**Fig 3. Enhanced breathing of tail-less nucleosomes.** Two-dimensional histograms depicting the conformational sampling of the 5' and 3' L-DNA arms in the space defined by the $\gamma_1$ and $\gamma_2$ angles. The arrow inserts in the lower-right corner indicate the direction of the opening, whereas the numbers are the areas of the histograms. The crosses indicate the most populated region of each histogram. The data for the 1 $\mu$s tail-less simulations is depicted in red, whereas the sampling of the complete nucleosome is showed in black.

different simulations and transitions between configurations occurred in the individual 1 $\mu$s simulations. Moreover, specific tail configurations associate with particular conformations of the nucleosomal DNA. When the H3 tail was in an intermediate position between the L-DNA and the dyad, it acted as a bridge keeping the nucleosome closed. When the tail had a large number of contacts to the outer gyre, and a short distance to the L-DNA, the nucleosomal DNA was partially closed. When the H3 or H2AC tails were near the dyad and away from the L-DNA forming only few contacts to the outer gyre, the nucleosomes opened.

The Esrrb$^{hH}$ nucleosome opened (750–850 ns of simulations 1) when $\delta$ between the H3 tail and the L-DNA as well as the $N_C$ between the tail with the dyad region increased (Fig 4A). At the same time, the H2AC tail had no contacts with L-DNA while its position relative to the dyad remained stable. (Fig 4B). Towards the end of the simulation, these contacts were quickly reformed, closing the DNA. Therefore, the collapse of the H3 tail on the DNA around the dyad together with the loss of the contacts of both tails with L-DNA permitted a large nucleosome opening (Fig 4A and 4B).

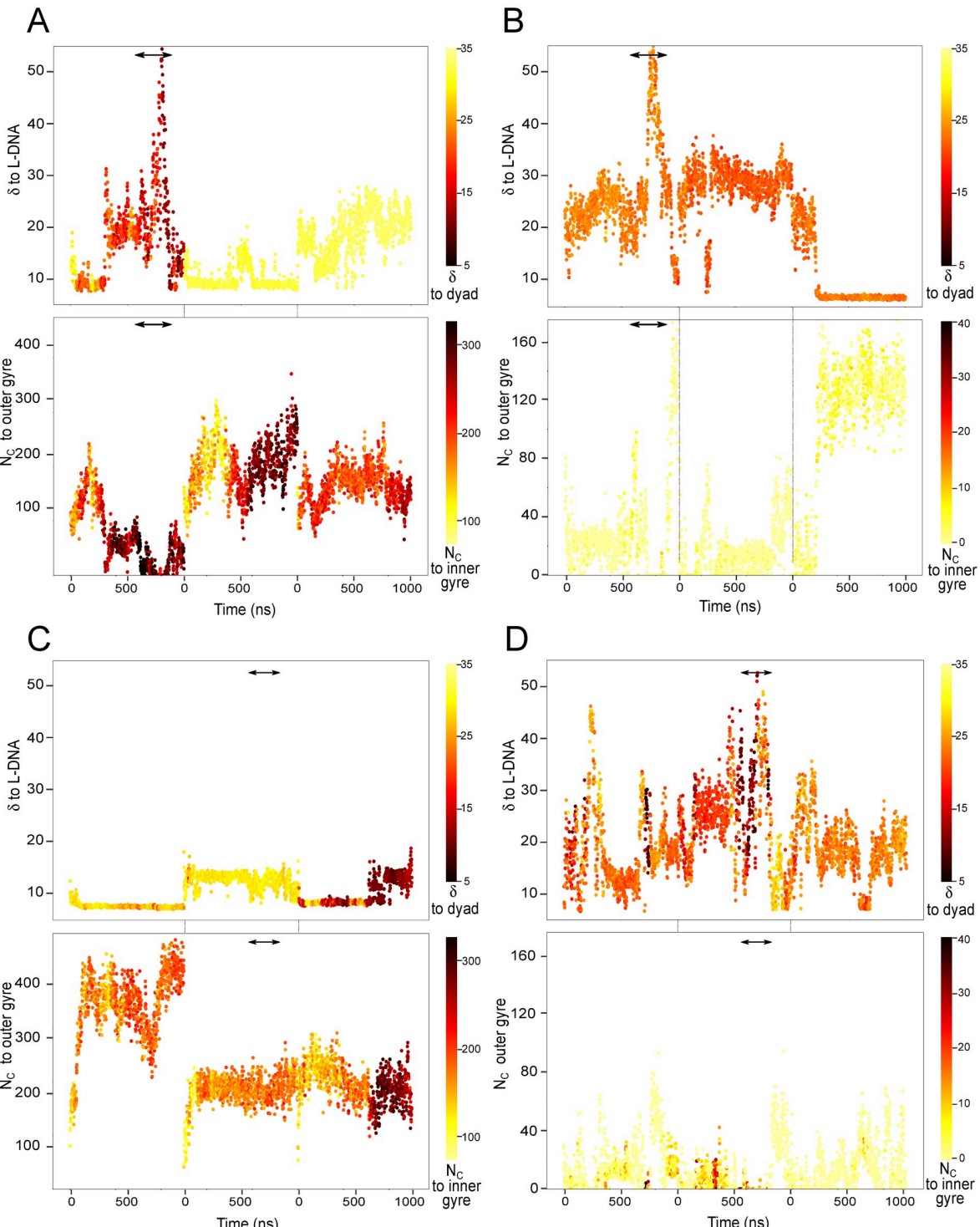

**Fig 4. Configurations of the H3 and H2AC tails.** Each panel shows the weighted mean distance ($\delta$) to the L-DNA (top) and the number of contacts ($N_C$) with the outer gyre (bottom) during the simulations. The points are colored by the $\delta$ to the dyad and the $N_C$ with the inner gyre of the DNA (see Methods for details on the definition of inner and outer gyre). (A-B) H3 (A) and H2AC (B) tails at the 3' end in the simulations of the Esrrb^{hH} nucleosome. (C-D) H3 (C) and H2AC (D) tails at the 5' end in the simulations of the Lin28b^{dH} nucleosome (time series in S2 Data).

The Lin28b$^{dH}$ nucleosome opened (700–800 ns, simulation 2), when the H3 tail collapsed around the middle of the 5' L-DNA, far from the dyad (Fig 4C). Before the opening, the configuration of the H2AC changed: its $\delta$ to the L-DNA increased while the $N_C$ with the outer gyre decreased(Fig 4D). Again, the contacts with DNA reformed shortly after opening, closing the DNA.

In conclusion, the H3 and H2AC tails sampled a broad range of configurations, but only an interplay between specific positions and conformations of both tails facilitated nucleosome breathing.

## Open and closed nucleosome conformations display specific histone tail configurations

Next, we tested whether there is a systematic correlation between the H3 and H2AC tail configurations and the nucleosome conformations. We split the hH and dH simulations in 2 separate groups and clustered the snapshots from each group in 8 clusters ranked by size in 3 different ways. First, to separate open and closed nucleosome conformations, we clustered based on the RMSD of the DNA backbone in the inner gyre and the outer gyre that opens (DNA clusters) (see Methods). Second, to separate histone tail configurations, we clustered based on the RMSD of the non-hydrogen atoms of the H3 and H2A histones separately (excluding the N terminal tail of H2A) (H3 and H2AC clusters).

Then, we mapped the DNA clusters on the 2D histograms of the $\gamma$ angles (Fig 5A and 5B and S4 Fig) and confirmed that they spread across the sampled conformational space of the nucleosomes. The distributions of the DNA clusters from the simulations of hH and dH nucleosomes were symmetric (Fig 5A and 5B and S4 Fig), demonstrating that they provide a similar phase space sampling. The largest DNA clusters ($DNA^{1h}$, $DNA^{1d}$) contained the more closed conformations in both simulation groups, whereas the smaller clusters ($DNA^{7h}$, $DNA^{8h}$ and $DNA^{6d}$) contained conformations open in both planes defined by $\gamma_1$ and $\gamma_2$.

To evaluate how DNA conformations associate with H3 and H2AC configurations, we computed the percentage of frames of each DNA cluster that is found in each H3 and H2AC cluster (Fig 5C and 5D). We characterized the histone tail configurations with the $R_g$ of the tails and the number of contacts with the inner and outer gyre of the nucleosome (Fig 5E and 5F).

The most open conformations of the DNA displayed specific configurations of both H3 and H2AC tails (Fig 5C and 5D). In the hH simulations, the open DNA conformations from cluster $DNA^{7h}$ (Fig 6A) and $DNA^{8h}$ had tail configurations from clusters $hH3^6$, and $hH2AC^4$ (Fig 5C) with few contacts with the outer DNA gyre, a wide distribution of conformations (wide $R_g$ distribution) and the most extended tail conformations sampled (Fig 5E). In the dH simulations, the open DNA conformations in cluster $DNA^{6d}$ (Fig 6B) had H3 configurations from cluster $dH3^5$, and often H2AC configurations from cluster $dH2AC^3$ (Fig 5D). Again, $dH3^5$ contains configurations with few contacts with both inner and outer DNA gyres and extended conformations (higher $R_g$) (Fig 5F). The dH2AC tail configurations are similar between different clusters because dH2AC is shorter and less positively charged compared to hH2AC (S2 Document). $dH2AC^3$ contains configurations with a slightly higher number of contacts with the inner DNA gyre. Therefore, the large opening of both genomic nucleosomes occurred when the H3 and H2AC tails formed few interactions with the L-DNA. We further confirmed these findings when we observed extensive opening of both the Esrrb$^{hH}$ and Lin28b$^{dH}$ nucleosomes in additional simulations started from closed nucleosomes with H3 and H2AC configurations selected from the clusters containing open nucleosome conformations (S1 Methods, S1 Text and S5 Fig).

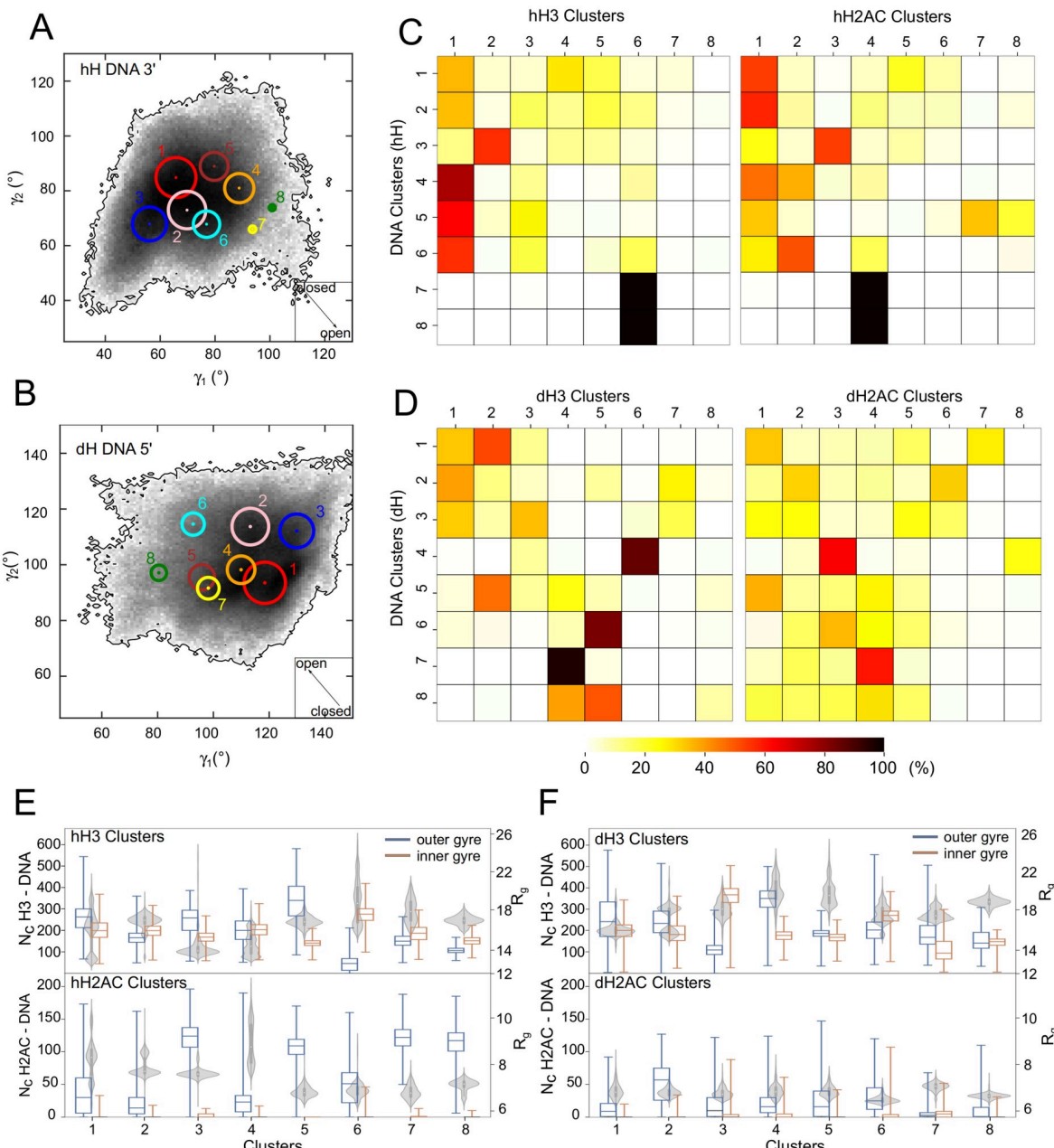

**Fig 5. Clustering of the DNA and histone tails.** (A-B) Position of the DNA cluster centroids in the 2D histogram of the $\gamma$ distribution for the 3' L-DNA in hH simulations (A) and 5' L-DNA in dH simulations (B). The size of the circles indicates the size of the clusters. (C-D) Distribution of the frames from each DNA cluster on the tail clusters (H3 in the left matrix, H2AC in the right), for hH (C) and dH (D) simulations. For each DNA cluster (row), the cells are colored by the percentage of frames of that cluster that belong to the tail cluster (column). (E-F) Features of H3 and H2AC configurations per cluster in the hH (E) and dH simulations (F). Box plots indicate the number of contacts between H3 (top) or H2AC (bottom) tail and the inner (red) and outer gyre (blue) of DNA per tail cluster. Violin plots represent $R_g$ of the H3 (top) or H2AC (bottom) tail per cluster.

The DNA conformations open only in one direction (defined by one $\gamma$) also displayed specific histone tail configurations. In the hH simulations, the DNA conformations open only in the direction defined by $\gamma_1$ from clusters $DNA^{4h}$ and $DNA^{5h}$ (Fig 6C) had almost exclusively H3 tail configurations from clusters $hH3^1$ and $hH3^3$ (Fig 5C) with a high number of contacts

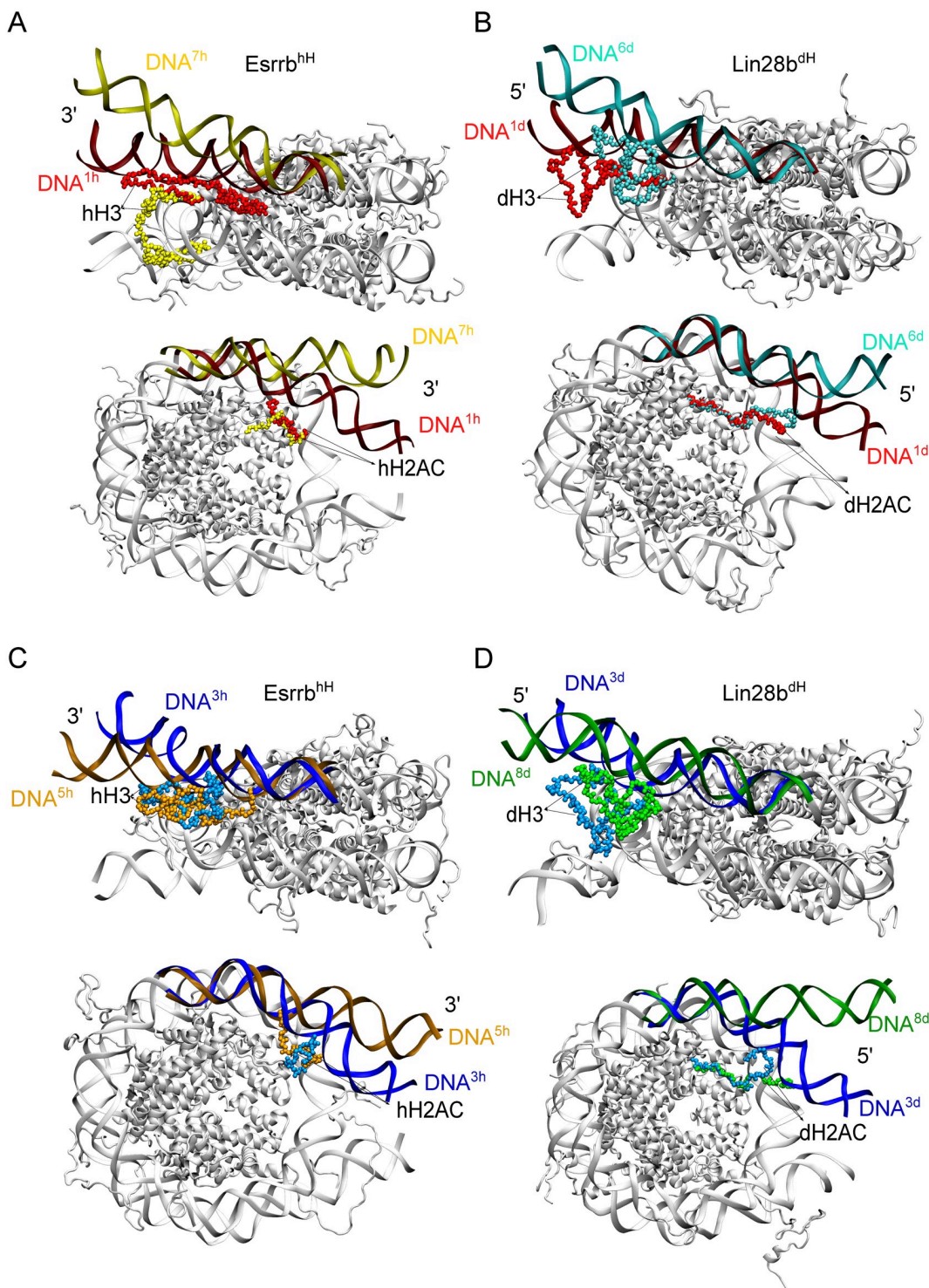

**Fig 6. Structures of DNA cluster representatives.** The L-DNA used for clustering is colored by the cluster color from Fig 5A and 5B. The L-DNA is in a dark shade, whereas the nearby histone tail is in a lighter shade of the same color. Each panel shows a side view of the nucleosome with the H3 tail (top), and a top view of the nucleosome with the H2AC tail (bottom). (A-B) Conformations closed or open in both the XZ ($\gamma_1$) and XY planes ($\gamma_2$). For the hH simulations (A), clusters $DNA^{1h}$ (closed) and $DNA^{7h}$ (open). For the dH simulations (B), clusters $DNA^{1d}$ (closed) and $DNA^{6d}$ (open). (C-D) Conformations open only in one plane, closed in the other. For the hH simulations (C), clusters $DNA^{3h}$ (open in the XY plane) and $DNA^{5h}$ (open in the XZ plane). For the dH simulations (D), clusters $DNA^{3d}$ (open in the XY plane) and $DNA^{8d}$ (open in the XZ plane). For clarity, the 3' end of the hH nucleosomes is shown in the same direction as the 5' end of the dH nucleosomes (views are inverted by 180˚ around the dyad axis).

to the outer DNA gyre, and compact conformations (low $R_g$ values) (Fig 5E). Therefore, when the nucleosome is closed in the XY plane ($\gamma_2$), the H3 tail adopts a compact, bridging configuration that interacts with both DNA gyres (Fig 5E). The DNA conformations open only in the XZ plane ($\gamma_1$) had H2AC configurations from the largest cluster, $hH2AC^1$ (Fig 5C) with less contacts to the outer DNA gyre (Fig 5E). The differences between the DNA conformations in clusters $DNA^{4h}$ and $DNA^{5h}$ are due to their different hH2AC tail configurations found in clusters $hH2AC^2$ and $hH2AC^7$-$hH2AC^8$ (Fig 5C).

The hH nucleosome conformations open only in the XY plane ($\gamma_2$) ($DNA^{3h}$, Fig 6C) had histone tail configurations from clusters $hH3^2$ and $hH2AC^3$ (Fig 5C) with extended conformations (Fig 5E). The H3 tail configurations in cluster $hH3^2$ do not vary in their position, displaying a narrow distribution of the number of contacts with both DNA gyres (Fig 5E).

The dH nucleosomes with conformations open only in the XZ plane ($\gamma_1$) in cluster $DNA^{8d}$ (Fig 6D) had H3 configurations from clusters $dH3^4$ and $dH3^5$, and variable H2AC configurations. $dH3^5$ contains tail configurations found in the most open nucleosome conformations. $dH3^4$ contains configurations with extended conformations and more contacts with the outer DNA gyre than those in $dH3^5$. These are present in the slightly more closed DNA conformations from cluster $DNA^{7d}$ (Fig 5D). Therefore, the conformations in $DNA^{8d}$ can be seen as transient between the more closed conformations in $DNA^{7d}$ and the most open conformations in $DNA^{6d}$. The dH nucleosome conformations open only on the XY plane ($\gamma_2$) in cluster $DNA^{3d}$ (Fig 6D) had mostly H3 configurations from clusters $dH3^1$ and $dH3^3$ and variable dH2AC configurations (Fig 5D). The $dH3^3$ has more extended dH3 tail conformations with more contacts with the core nucleosomal DNA (Fig 5F). The $dH3^1$ cluster is the largest dH3 cluster with more diverse dH3 configurations (Fig 5F). These findings further confirm that the loss of a large fraction of the contacts between H3 and H2AC tails and the L-DNAs is required for nucleosome opening.

The most closed DNA conformations in the clusters $DNA^{1h}$, $DNA^{1d}$ (Fig 6A) display H3 tail configurations found in the largest H3 clusters or distributed among different clusters (Fig 5C and 5D) that are either extended or more compact and have a large number of contacts both with L-DNAs and the core DNA (Fig 5E and 5F). Therefore, the nucleosomes are maintained in a closed conformation by H3 tail mediated bridging interactions between the L-DNA and the core nucleosomal DNA. The closed conformations of the hH nucleosome have H2AC tail configurations with extended conformations and a moderate number of contacts with the L-DNA (Fig 5C and 5E). In contrast, those of the dH nucleosomes do not display any specific H2AC tail configurations (Fig 5D and 5F).

The clustering analysis demonstrated that the open and closed nucleosome conformations are characterized by specific H3 and H2AC tail conformations and positions. In particular open nucleosome conformations have H3 and H2AC tail configurations with fewer contacts with the DNA, especially with the adjacent L-DNA.

## Nucleosome opening is induced by H3 and H2AC tail dynamics

To test whether the motions of the H3 and H2AC tails are a consequence of nucleosome breathing or induce it, we mapped the structural snapshots from the DNA, H3, and H2AC clusters on the simulation time traces, and compared the distributions with the nucleosome breathing profiles given by the time evolution of the $R_g$ (Fig 7).

We found that reversible conformational transitions in the nucleosomal DNA are frequent in 1 $\mu$s, while transitions to largely open conformations are rare. In the hH simulations, the most open DNA conformations from clusters $DNA^{7h}$ and $DNA^{8h}$ are sampled only in the second simulation of the Esrrb$^{hH}$ nucleosome between 700 and 800 ns (Fig 7A). In the dH

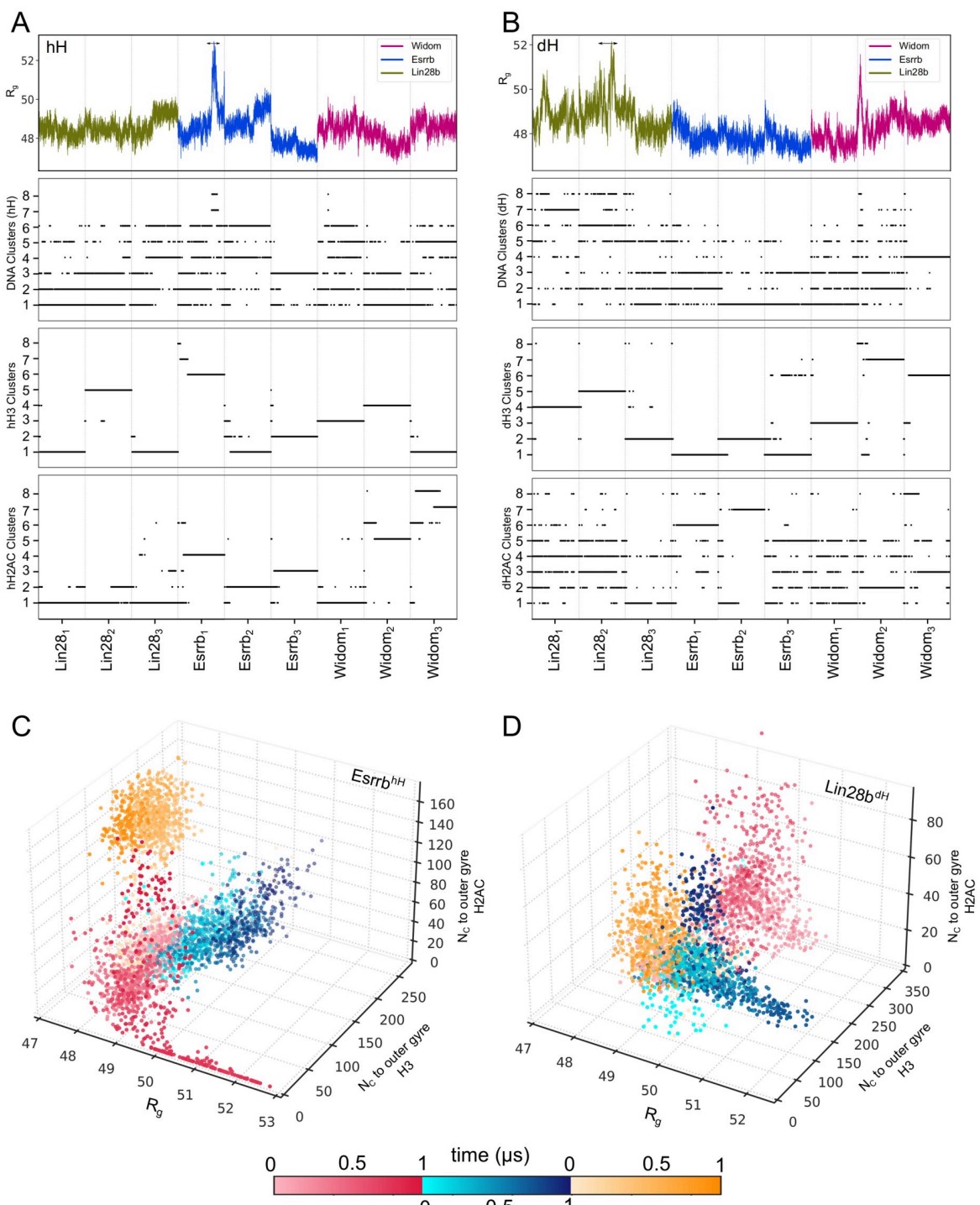

**Fig 7. Correlated structural transitions in the nucleosomes.** Structures from each DNA, H3, and H2AC cluster mapped on the simulation time series. (A) The 3' end of the hH nucleosomes. (B) The 5' end of the dH nucleosomes. In each panel, the top plot shows the nucleosome $R_g$ and the 3 plots below show to which DNA, H3, and H2AC clusters each simulation snapshot belongs to. Each point corresponds to one single snapshot. (C-D) Scatter 3D plots depicting the correlation between the conformations of Esrrb$^{hH}$ (C) and Lin28b$^{dH}$ (D) and the number of contacts of the H3 and H2AC tails with the outer DNA gyre during the individual simulations (red, blue, yellow). The light to dark color palette shows the simulation time.

simulations, they are in clusters $DNA^{6d}$ and $DNA^{8d}$ and are sampled mainly in the first and second simulations of the Lin28b$^{dH}$ (Fig 7B). The opening of Lin28b$^{dH}$ was more gradual than that of Esrrb$^{hH}$.

In contrast, transitions between configurations of the H3 tail were rare in 1 $\mu$s (Fig 7), different configurations being sampled in the independent simulations. This highlights the importance of grouping the independent simulations in ensembles for investigating such large scale motions. Transitions between H2AC tail configurations in 1 $\mu$s were more frequent, especially for the shorter dH2AC.

Both large openings of the nucleosomal DNA were preceded by transitions of both H3 and H2AC tail configurations. Esrrb$^{hH}$, opened after both H3 and H2AC tails adopted specific conformations from clusters $hH3^6$ and $hH2AC^4$ (Fig 7A). Lin28b$^{dH}$ opened when the H3 tail was in a configuration from the $dH3^5$ cluster and the H2AC tail transited to configurations from clusters $dH2AC^6$ and $dH2AC^8$ before the nucleosome opened. The loss of contacts between the tails and the outer DNA gyre precedes the opening of both Esrrb$^{hH}$ (Fig 7C) and Lin28b$^{dH}$ (Fig 7D). The correlation between nucleosome opening and histone tail dynamics was also confirmed by a principal component analysis of the simulation ensembles (S1 Methods, S1 Text, and S6 Fig).

The interplay between both tails also contributed to a very short lived opening in the first simulation of the Widom$^{dH}$ nucleosome (Table 1, Fig 7B). Here, the H3 tail sampled configurations of the $dH3^3$ cluster with less contacts to the outer DNA gyre, and higher R$_g$ values indicating a more extended conformation (Fig 5F). However, because the dH2AC tail didn't adopt configurations permissive for DNA opening ($dH2AC^6$ and $dH2AC^8$) the nucleosome remained mostly is a very closed conformations.

In conclusion, the extensive opening of the nucleosomes developed only after both the H3 and H2AC tails adopted specific configurations lacking the interactions required to maintain the nucleosomes closed. Transition out of these configurations lead to nucleosome closing.

## Nucleosome opening is modulated by epigenetic regulatory residues

To test how interactions between aminoacids in the histone tails and the DNA impact on nucleosome conformational dynamics, we monitored the minimal distance between positively charged lysines and arginines and the inner and outer DNA gyres. (Fig 8). Post-translational modifications of these key residues mark active versus inactive chromatin, are involved in epigenetic regulation of gene expression, and are expected to impact on nucleosome dynamics. We separated the residues of the H3 tail in 4 groups: the "tip" group includes the residues at the tip of the tail R2, K4, the "center-tip" group includes residues R8, K9, K14, the "center-anchor" group includes residues R26 and K27, and the "anchor" group includes the residues anchoring the L-DNA to the inner gyre of the DNA K36, K37, R40 and R42. Both large opening events involved the loss of interactions between these residues with the outer DNA gyre.

In the open conformations of the Esrrb$^{hH}$ nucleosome the H3 tail configurations (cluster $hH3^6$) lacked interactions with the outer DNA gyre except for the hydrogen bonds formed by R42 from the anchor group. The interactions between the tip and center-tip groups and the outer DNA gyre were not formed in the simulation in which the nucleosome opened. The interactions of the center-anchor residues were lost before the nucleosome opened. Therefore, the loss of these interactions is a prerequisite for the opening (S7 Fig). Some of the residues from the tip and the center groups (R2, R8, R26) repositioned to interact with the inner gyre. Interestingly, most lysines involved in epigenetic regulation do not interact with DNA in these configurations (Fig 8A). The H2AC tail configurations in open nucleosomes (cluster $hH2AC^4$) do not interact with DNA (Fig 8B), the interactions breaking before the opening. In contrast,

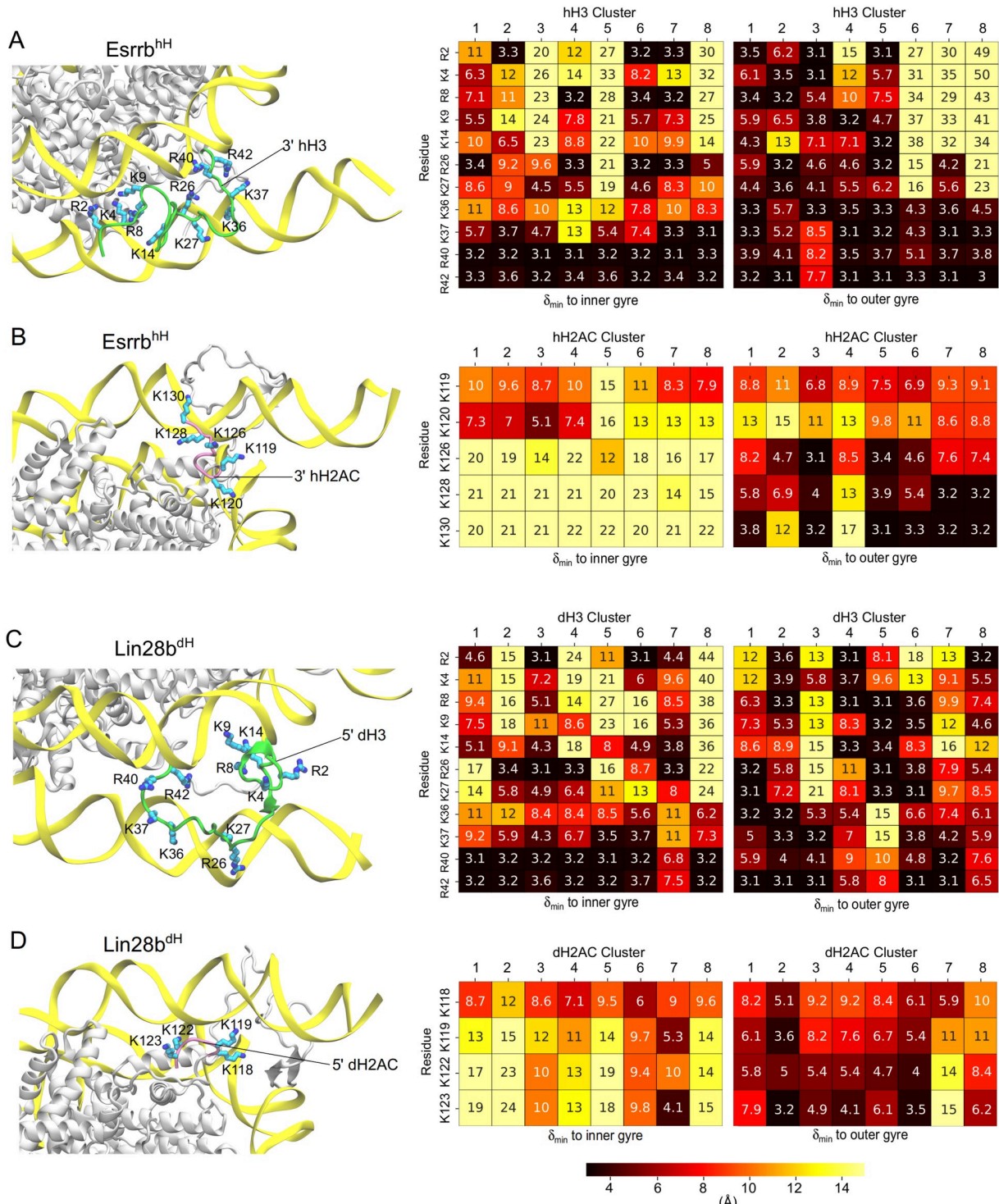

**Fig 8. Interactions of epigenetic regulatory residues from the histone tails with DNA.** Each panel shows a structural illustration of the histone tail (left) and 2 heatmaps with minimal distances of the analyzed residues to the inner (middle) and outer gyre of the DNA (right). (A-B) the hH simulations. (C-D) the dH simulations. The H3 (A,C) and H2AC (B,D) tails are shown as green and mauve ribbons with the analyzed aminoacids colored by atom name. The DNA is in yellow ribbons and the core histones are in white. The heatmaps show the analyzed residues on the vertical and the DNA clusters on the horizontal. The numbers are the median value of the minimal distance of each residue to the DNA in the corresponding DNA cluster.

in the closed conformations of Esrrb$^{hH}$ K126, K128, and K130 formed stable interactions with the outer DNA gyre (S8 Fig).

In the open conformations of the Lin28b$^{dH}$ nucleosome, the H3 tail configurations (cluster $dH3^5$) interacts with the outer DNA gyre with the residues in the center groups (R8, K9, K14, R26, K27) (Fig 8C). The residues of the tip and anchor groups do not interact with the DNA while some anchor residues (K37, R40, R42) interact with the inner DNA gyre (Fig 8C and S7 Fig). In addition, the interactions between the dH2AC tail and the outer gyre broke before the opening (Fig 8D and S8 Fig).

In the hH nucleosome conformations open only in one direction ($\gamma_1$ or $\gamma_2$) the H3 tails maintained some of its interactions with the outer DNA gyre. In the configurations from cluster $hH3^3$ found in nucleosome conformations open in the XZ plane ($\gamma_1$), the anchor group interacts with the inner DNA gyre (Fig 8A). On the other hand, the configurations from clusters $hH3^2$, $hH3^4$, and $hH3^5$ found in nucleosome conformations open in the XY plane ($\gamma_2$), the anchor group interacts with the outer DNA gyre, whereas the residues in the tip and center groups do not interact with DNA (Fig 8A).

In dH nucleosome conformations open in the XZ plane ($\gamma_1$), the H3 configurations (clusters $dH3^4$ and $dH3^5$) have no interactions between the anchor group and the outer DNA gyre. On the other hand, in the configurations from clusters $dH3^1$ and $dH3^3$ found in nucleosomes open in XY plane ($\gamma_2$), anchor but not tip and center residues interact with the outer gyre (Fig 8C).

In the closed nucleosome, the H3 and H2AC configurations have the most interactions with the outer DNA gyre. Therefore, the amplitude of nucleosome breathing depends on the number of interactions between aminoacids in the H3 and H2AC tails and the nearby L-DNA. When many of these are formed, the nucleosomes remained closed. For a large nucleosome opening, the loss of most but not all interactions between the tails and the L-DNA was required. The residues involved are located in clusters of positively charged residues and the interactions of one residue may be substituted by interactions of nearby residues from the same cluster. This suggests that these residues may be accessible for chemical modifications also in closed nucleosomes.

## Discussion

Chromatin fibers are not as compact as it has been thought [58]. Hence, their folding and dynamics are more sensitive to the ratio between open and closed nucleosomes. It becomes evident that elucidating the mechanisms by which nucleosomes open and close is key to understand chromatin dynamics.

Here we showed how the interplay between the histone H3 and H2AC tails controls the breathing of genomic nucleosomes. From a total of 24 $\mu$s of atomistic MD simulations, we observed 2 large opening events in 2 different genomic nucleosomes and only low amplitude breathing motions in an engineered nucleosome. Our findings suggest that there are lower barriers for large amplitude opening in genomic nucleosomes. This supports previous findings [39] that genomic nucleosomes are more flexible and mobile than artificial nucleosomes bearing strong positioning DNA sequences such as the Widom 601 sequence [40]. The simulations of tail-less nucleosomes confirmed that the nucleosome structural flexibility is encoded in the DNA sequence as the breathing pattern was similar in complete and tail-less nucleosomes.

Previously, we reported that the larger structural flexibility of the Lin28b nucleosome with *Drosophila* histones was in agreement with its lower thermal stability in experiments [39]. Lin28b opened extensively in the simulations with *Drosophila* histones and adopted more open conformations in all simulations, suggesting that it is the less compact nucleosome.

Here we also report a high structural flexibility leading to a large opening event in the Esrrb nucleosome with human histones in contrast to the Esrrb nucleosome with *Drosophila* histones which was more rigid. The Esrrb nucleosome had a lower thermal stability than the Widom nucleosome but higher than Lin28. Moreover, its stability varied in the experiment [39], suggesting that it may adopt both more or less stable conformations.

However, caution is needed when comparing the simulations with these experiments. Although both nucleosome disassembly in the experiments and nucleosome breathing in the simulations are measures of nucleosome structural flexibility, they differ in amplitude and time scale of the motions involved. We propose that the variability in our simulations is not due to the minor differences between the human and *Drosophila* histone sequences (S1 Document) but rather to the differential sampling of histone tail dynamics (see Discussion below).

Complete nucleosome unwrapping is thought to happen in seconds, with intermediate states forming after hundreds of milliseconds [59, 60]. While atomistic MD simulations are very powerful in in studying nucleosome dynamics, the timescales accessible are in the $\mu$s range [36, 37, 39, 61]. Therefore, the simulations are not converged and achieve only a limited sampling of the conformational space. This limitation may be circumvented by enhanced sampling techniques [61, 62]. However, these involve biasing the motions which may provide inaccurate estimations of the energies and pathways of nucleosome motions [61]. We argue that the grouping of multiple independent simulations started from the same structure (replicas) in ensembles is a powerful approach to achieve extensive sampling. In each replica, the nucleosome samples a subregion of the phase space and two replicas are never identical because of the randomizations performed at the beginning of each simulation. Therefore, merging the replicas expands the phase space explored. This approach enabled us to monitor the 2 rare large opening events in complete nucleosomes and study their mechanisms. The $\gamma_1$-$\gamma_2$ histograms demonstrated that the openings were extensions of the low amplitude nucleosome breathing. In contrast to previous reports of larger amplitude opening in atomistic simulations of incomplete nucleosomes, we could elucidate the role of histone tails in these motions [63].

Histone tails are important for chromatin dynamics. H3 and H4 tails impact the packing of the genome by modulating the interaction between nucleosomes [6, 7]. Moreover, the tails interfere with the binding of other proteins to the nucleosome. For example, the H2AC tail interacts with the linker histone impacting on its role to increase chromatin compaction upon binding to nucleosomes [8]. However, the mechanisms by which the tails impact on nucleosome structural flexibility remained obscure.

The role of the tails has been proposed from a number of coarse grained simulations [15, 18, 21, 23]. In these simulations, the histone tails are simplified and the detailed chemistry of the interactions involved is often neglected, leading to an incomplete representation of the protein-DNA interactions. Precisely this gap is closed by atomistic simulations. However, simulating large amplitude nucleosome motions and accurately describing the dynamics of histone tails is challenging. The simulations of intrinsically disordered proteins or protein regions like histone tails is particularly difficult, because the force-fields used for structured proteins often do not provide a correct sampling of their conformational space [64]. However, the force fields we used represent the state-of-the-art, and reproduce the dynamics of intrinsically disordered proteins with reasonable accuracy [64]). Force fields from this family have been used to explain the destabilization of the nucleosome by mutations in H2A in experiments, confirming their accuracy [65].

The atomistic studies to date have been performed using either incomplete nucleosome or nucleosomes with artificial sequences such as the Widom or the human $\alpha$-satellite DNA. Large scale breathing observed in tail-less nucleosomes provided insights into the interactions of the histone core with DNA. However, removing the histone tails in these simulations lead to unnatural breathing [25, 27–30].

Here, we report a regulation mechanism for breathing of complete, genomic nucleosomes that involves an interplay between the histone H3 and H2AC tails and specific interactions between residues in these tails and the DNA. The general behavior of the H3 and H2AC tails was similar in the simulations with *Drosophila* and human histones. The tails collapsed on the DNA and large changes in the tail conformation, position, and DNA binding pattern were rare in 1 $\mu$s. However, the tail configurations were different in the simulations of the same ensemble, allowing a more extensive sampling. The tail-less nucleosomes opened more extensively indicating that the histone tails maintain nucleosome closed. Becasue H3 and H2AC tails are the only tails near the L-DNA arms, we propose that nucleosome opening and closing is a direct result of cooperative motions of these two tails. Our findings explain previous reports that the deletion of either of the 2 tails lead to large amplitude nucleosome opening in experiments [8, 66, 67] and in high temperature simulations [38].

Post-translational, chemical modifications of key residues in histone tails mark chromatin regions as active or inactive, establishing the epigenetic regulation of gene expression at a given time and cellular context. Therefore, it is of utmost importance to understand the role of these residues in nucleosome and chromatin dynamics. We show that epigenetically active residues in the H3 tail are located in clusters of positively charged residues that interact with DNA. We observed that for a large opening of nucleosomes, the majority of these interactions, especially those with the L-DNA have to be absent. Our findings suggest that the nucleosome is kept closed when a majority of these interactions are formed. However, the absence of interactions of each individual residue may be compensated by interactions of residues from the same cluster, suggesting that these residues may be available for chemical modification even in closed nucleosomes. For example, K9 and K27, both methylated in inactive chromatin did not establish specific interaction with DNA independent in any nucleosome conformation. However, arginines flanking them displayed a nucleosome conformation dependent pattern of interactions with DNA. K36 from the region anchoring L-DNA to the core DNA is acetylated in active chromatin and formed interactions with the DNA only in closed nucleosomes, suggesting that this residue is more accessible in open nucleosomes. However, K36 is flanked by 2 arginines and another lysine forming redundant interactions with the DNA. Therefore, our findings suggest the tail residues are accessible for chemical modification independent of nucleosome conformation. The impact of such modifications on intra-nucleosome motions may manifest in more subtle ways than just by neutralizing the positive charge of the tail residues. Studying how single modifications or histone variants affect nucleosome dynamics with atomistic MD simulations is a powerful alternative to experiments [37, 62, 68, 69]. We predict that with the continuous increase of computational resources available, long simulations of complete genomic nucleosomes, modified or unmodified will contribute to elucidating these mechanisms.

## Conclusions

Here we demonstrated how the interplay between the histone tails modulates the nucleosome conformational dynamics. Large opening of nucleosomes is possible only when the H3 and H2AC tail adopt specific configurations that involve only few contacts with the linker DNA arms of the nucleosome. In addition, we showed that such large openings are more frequent in genomic nucleosomes compared with engineered nucleosomes with DNA sequences optimized for strong positioning such as the Widom sequence. Finally, we harvested the potential of atomistic molecular dynamic simulations to reveal nucleosome and histone tail dynamics and mechanistic insights on how these are interconnected. Our findings are based on unprecedented sampling of nucleosome dynamics in atomistic simulations from which we revealed

two large amplitude opening events of two genomic nucleosomes. Future simulations of larger amplitude motions of single or connected nucleosomes on longer timescales will be needed to describe events such as ample nucleosome breathing, nucleosome repositioning or disassembly. From such simulations, the impact of the interplay between histone tails and linker DNA in modulating intra- and inter-nucleosomes interactions as well as how this triggers changes in the chromatin state may be inferred.

## Supporting information

**S1 Methods. Supplementary methods.**
(PDF)

**S1 Text. Overview and supporting results.**
(PDF)

**S1 Document. Alignment of the *Drosophila and human histones*.**
(PDF)

**S2 Document. DNA sequences in the Widom, Esrrb and Lin28B nucleosomes.**
(PDF)

**S1 Data. Complete time series 1.** Spreadsheets (bzip2 compressed Excel format) with the time series (every 40 ps) of $R_g$ and $\gamma$ angles. The document has 2 sheets titled by the properties they contain. Each column contains the data from one simulation ensemble with the individual simulations following each other in rows. Each row represents a simulation frame.
(BZ2)

**S2 Data. Complete time series 2.** Spreadsheets (bzip2 compressed Excel format) with the time series (every 100 ps) of the median interatomic distances and the number of contacts between the H3 and H2AC tails and different DNA segments. The document has 4 sheets titled by the properties and simulations (dH or hH) they contain. Each column contains the data from one simulation ensemble with the individual simulations following each other in rows. Each row represents a simulation frame.
(BZ2)

**S1 Fig. Number of histone-DNA contacts.** The 2 panels depict the time evolution of the number of contacts of the histone core (top) and histone tails (middle and bottom) to the DNA. For the histone tails, contacts were split between contacts to the inner gyre (middle) and contacts to the outer gyre (bottom). (A) Contact evolution in the hH simulations. (B) Contact evolution in the dH simulations. A contact was defined as a non-hydrogem atom closer than 4.5 Å to another non-hydrogen atom.
(PDF)

**S2 Fig. Nucleosome structural flexibility.** (A-B): The evolution of the $R_g$ of the DNA over time, during the hH (A) and dH (B) simulations. (C-F) Time series and histograms of the $\gamma$ angles described in Fig 2, for the 5' arm of the hH nucleosomes (C), 5' arm of the dH nucleosomes (D), 3' arm of the hH nucleosomes (E) and 3' arm of the dH nucleosomes (F). The individual simulations are separated by vertical black lines.
(PDF)

**S3 Fig. Radius of gyration of the histone tails.** The evolution of the $R_g$ over time for histones H3 and H2AC tails. (A) hH simulations. (B) dH simulations.
(PDF)

**S4 Fig. Sampling of the DNA clusters.** Two-dimensional histograms depicting the conformational sampling of the L-DNA arms in the space defined by the $\gamma_1$ and $\gamma_2$ angles for each individual DNA cluster. (A) 3' L-DNA clusters from the hH simulations. (B) 5' L-DNA clusters from the dH simulations.
(PDF)

**S5 Fig. Nucleosome opening in simulations with selected nucleosome and histone tail conformations.** (A) Table summarizing the simulations we performed to probe the reproducibility of the extensive nucleosome opening (similar to main Table 1). (B-C) Two-dimensional histograms depicting the conformational sampling of the L-DNA arms in the space defined by the $\gamma_1$ and $\gamma_2$ angles for the Esrrb$^{hH}$ (B) and Lin28b$^{dH}$ (C) nucleosomes. In black are the original histograms (see Fig 2), in green the combined sampling of the three simulations started with a closed nucleosome but with H3 and H2AC tails in configurations found in open nucleosomes. Each simulation is depicted by blue, yellow, and red contours. (D-E) Time series for the $R_g$ during the three 0.5 $\mu$s simulations of Esrrb$^{hH}$ (D) and Lin28b$^{dH}$ (E).
(PDF)

**S6 Fig. Correlated motions of the histone tails and linker DNA.** (A-D) Superposition of DNA and H3 (A-B) or DNA and H2AC (C-D) snapshots from the pseudotrajectory of the lowest frequency principal component (PC1) of the simulation ensembles of Esrrb$^{hH}$ (A,C) and Lin28b$^{dH}$ (B,D). (E-F) The $R_g$ along PC1 pseudotrajectory of Esrrb$^{hH}$ (E) and Lin28b$^{dH}$ (F). The closed to open transition is indicated by the Red-White-Blue color scale. The amplitude was normalized such as -1 corresponds to the most closed and +1 to the most open conformation.
(PDF)

**S7 Fig. Interactions of H3 residues with DNA.** The evolution of the H3 residues position relative to the inner and outer gyre of the DNA. (A) The Esrrb$^{hH}$ nucleosome. (B) The Lin28b$^{dH}$ nucleosome. The plot at the top row shows the nucleosome $R_g$ to monitor opening and closing events. The other plots show the minimal distance of the residues to the outer gyre, colored by the minimal distance to the inner gyre of the DNA.
(PDF)

**S8 Fig. Interactions of H2AC residues with DNA in the nucleosomes with large opening.** The evolution of the H2AC residues position relative to the inner and outer gyre. (A) The Esrrb$^{hH}$ nucleosome. (B) The Lin28b$^{dH}$ nucleosome. The plot at the top row shows the nucleosome radius of gyration to monitor opening and closing events. The other plots show the minimal distance of the residues to the outer gyre, colored by the minimal distance to the inner gyre of the DNA.
(PDF)

## Acknowledgments

The authors thank Caitlin M MacCarthy for the support and discussion. J.H. is part of the International Max Planck Research School-Molecular Biomedicine, Münster, Germany.

## Author Contributions

**Conceptualization:** Jan Huertas, Vlad Cojocaru.

**Data curation:** Jan Huertas, Vlad Cojocaru.

**Formal analysis:** Jan Huertas, Vlad Cojocaru.

**Funding acquisition:** Jan Huertas, Hans Robert Schöler, Vlad Cojocaru.

**Investigation:** Jan Huertas, Vlad Cojocaru.

**Methodology:** Jan Huertas, Vlad Cojocaru.

**Project administration:** Hans Robert Schöler, Vlad Cojocaru.

**Resources:** Jan Huertas, Hans Robert Schöler, Vlad Cojocaru.

**Supervision:** Vlad Cojocaru.

**Validation:** Jan Huertas, Vlad Cojocaru.

**Visualization:** Jan Huertas, Vlad Cojocaru.

**Writing – original draft:** Jan Huertas.

**Writing – review & editing:** Jan Huertas, Hans Robert Schöler, Vlad Cojocaru.

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
