## [Decision Letter · Decision Letter 0]

7 Jan 2021

Dear Dr. Cojocaru,

Thank you very much for submitting your manuscript "Histone tails cooperate to control the breathing of genomic nucleosomes" for consideration at PLOS Computational Biology.

As with all papers reviewed by the journal, your manuscript was reviewed by members of the editorial board and by several independent reviewers. In light of the reviews (below this email), we would like to invite the resubmission of a significantly-revised version that takes into account the reviewers' comments.

We cannot make any decision about publication until we have seen the revised manuscript and your response to the reviewers' comments. Your revised manuscript is also likely to be sent to reviewers for further evaluation.

Sincerely,

Alexander MacKerell

Associate Editor

PLOS Computational Biology

Arne Elofsson

Deputy Editor

PLOS Computational Biology

Reviewer's Responses to Questions

**Comments to the Authors:**

Reviewer #1: Authors investigated the roles of two histone tails in large scale opening motion (breathing) of nucleosomes using powerful atomistic molecular dynamics (MD) simulations. Three 1-microsecond long simulations of nucleosomes for two genomic DNAs with the two histone sets showed rare and large scale openings in a few events, while those for the 601 nucleosome did not show such a large breathing motion. Specific roles of the N-terminal tail of H3 and the C-terminal tail of H2A are carefully analyzed.

Strength: I consider it one of the most comprehensive atomistic MD works on nucleosomes to date, making it a very important contribution to the field. Personally, I learned a lot from the result and thus like this work very much. I think the conclusion make sense perfectly.

Concern: However, due to its intrinsic hardness of the sampling, authors could observe only one event in Esrrb nucleosome and ~two(?) events in Lin28b nucleosome. I have a concern on the statistics and the robustness of the results, as will be explained below. Related to it, I would suggest one additional simulation to strengthen the paper.

The specific comments:

1) Even though authors performed, in total, 18 microsecond MD runs, they found one large-scale opening event in Esrrb nucleosome and two? in Lin28b nucleosome. Based on the fact that this event is preceded by selected conformations of the tails, authors tried to conclude that "Transitions between open and closed

nucleosome conformations were driven by the displacement and changes in compaction

of the two histone tails."(from abstract, for example).

However, looking at Fig. 6, for the Esrrb case, there is only one event only in 3' end linker DNA, at which the H3 tail was in the 6-th cluster. What does this mean? This can be just by chance, since this is the only one event. (I myself do not think it by chance, though) Unfortunately, we cannot compare it with the case of Lin28b because the clustering was done separately(which is unavoidable, though). Therefore, as it is, the appeal is simply way too strong.

Related to the point, I would suggest an additional small MD runs. If the authors' view is correct, specific conformations of H3 and H2A C tails facilitate the large-scale opening. In a separate MD setup, they can restrain these tails near the conformations found in 700-800 ns of the first run of the Esrrb nucleosome, and conduct additional runs of 1 microseconds. If they find large-scale opening repeatedly, this markedly strengthens their appeal.

2) I thought the current way of presenting the result unnecessarily complicated. This may be partly because of the numbering in the cluster analysis. Currently, authors sorted 8 clusters, probably, by their sizes, which is automatic, but does not help our understanding much. I would suggest the ordering by some physical parameter. For example, the DNA clusters can be sorted by the gamma_1-gamma_2 in 5' end and gamma_2-gamma_1 in 3' end, which is much more intuitive. The other two clusters for the tails can be sorted based on the data in Fig. 7 (Differences in the distances of R2-K14 between inner and outer gyres, for example).

3) I really eager to see one nice scattered plot that directly indicated the correlation between the tail conformations and the degree of opening. For the latter, one can use gamma_1 - gamma_2 as a simple score. Or, RoG itself is a nice score. For the tail conformations, I think, again, that the data in Fig. 7 can be rearranged to make one score (an order parameter) that monitors if the tail is directed to the outer gyre/linker DNA or to the dyad region. With the two scores in x- and y-axis, one can make the scattered plot showing their correlation, I think.

4) I find it surprising that the nucleosome dynamics is rather different between hH and dH. Can authors argue why so? What part of difference caused this difference?

Reviewer #2: Chromatin remodeling is cucial for proper expression of the genetic information in eukaryotes, and opening of the nucleosome is an important step in this process, which is however difficult to study experimentally. Here extensive all-atom molecular dynamics simulations are used to shed light on the role of interactions between histone tails and nucleosomal DNA in the opening of the nucleosome. The main result of the study is that there does indeed seem to be a relation between tail conformations of two of the histones and conformations of the DNA when there is an opening motion in the nucleosome.

As the authors note, the simulations are however too short (3 x 1 microsecond for each system) to be converged wrt to nucleosome opening; opening events of short (100 ns) duration are only observed in two out of a total of 18 simulations. It is therefore not really possible to draw any conclusions regarding differences between the DNA sequences used in terms of their ability to support opening events.

Three different DNA sequences have been used, two "genomic" and one "engineered". The rationale for choosing these sequences is not given in the manuscript, nor is the difference of the "engineered" sequence from the other two described; in particular it would be of interest to know if this DNA differs in its ability to interact with the relevant histone tails. The results in figure 2 do not convincingly show any significant/important differences in the behavior between the different systems. Is the area of the distributions really an important observable, in particular since the peripheral regions of the distributions are not very highly populated?

The discussion of epigenetic histone modifications is relevant, and it would strengthen the manuscript to have simulation data for nucleosomes with either or both of the histones of interest modified.

Is the radius of gyration wrt to an axis or wrt the center of mass? Please also note that it is usually denoted Rg, where the g is a subscript - introducing a new symbol "RoG" is just confusing.

Since three replica have been simulated for each system it should be possible to provide error estimates for quantities of interest.

**Have all data underlying the figures and results presented in the manuscript been provided?**

Reviewer #1: Yes

Reviewer #2: None

PLOS authors have the option to publish the peer review history of their article (what does this mean?). If published, this will include your full peer review and any attached files.

Reviewer #1: No

Reviewer #2: No
---

## [Decision Letter · Decision Letter 1]

28 Apr 2021

Dear Dr. Cojocaru,

We are pleased to inform you that your manuscript 'Histone tails cooperate to control the breathing of genomic nucleosomes' has been provisionally accepted for publication in PLOS Computational Biology.

Best regards,

Alexander MacKerell

Associate Editor

PLOS Computational Biology

Arne Elofsson

Deputy Editor

PLOS Computational Biology

Reviewer's Responses to Questions

**Comments to the Authors:**

Reviewer #1: Authors addressed all the previous comments appropriately and I consider that the revised manuscript is ready to be published.

Reviewer #2: I am satisfied with the revised version.

**Have the authors made all data and (if applicable) computational code underlying the findings in their manuscript fully available?**

Reviewer #1: Yes

Reviewer #2: Yes

PLOS authors have the option to publish the peer review history of their article (what does this mean?). If published, this will include your full peer review and any attached files.

Reviewer #1: No

Reviewer #2: No

---

## [Editor Report · Acceptance letter]

12 May 2021

PCOMPBIOL-D-20-02236R1 

Histone tails cooperate to control the breathing of genomic nucleosomes

Dear Dr Cojocaru,

I am pleased to inform you that your manuscript has been formally accepted for publication in PLOS Computational Biology. Your manuscript is now with our production department and you will be notified of the publication date in due course.

With kind regards,

Kata Acsay
